# MASTERING MASSIVE MULTI-TASK REINFORCEMENT LEARNING VIA MIXTURE-OF-EXPERT DECISION TRANSFORMER

**Yilun Kong[1], Guozheng Ma[2], Qi Zhao[1], Haoyu Wang[1],**
**Li Shen[3]\*, Xueqian Wang[1]\*, Dacheng Tao[2]**
[1]Tsinghua University; [2]Nanyang Technological University; [3]Sun Yat-sen University
`{kyl22,zhaoqi24,haoyu-wa22}@mails.tsinghua.edu.cn;`
`GUOZHENG001@e.ntu.edu.sg; mathshenli@gmail.com;`
`wang.xq@sz.tsinghua.edu.cn; dacheng.tao@ntu.edu.sg`

## ABSTRACT

Despite recent advancements in offline multi-task reinforcement learning (MTRL) have harnessed the powerful capabilities of the Transformer architecture, most approaches focus on a limited number of tasks, with scaling to extremely massive tasks remaining a formidable challenge. In this paper, we first revisit the key impact of task numbers on current MTRL method, and further reveal that naively expanding the parameters proves insufficient to counteract the performance degradation as the number of tasks escalates. Building upon these insights, we propose M3DT, a novel mixture-of-experts (MoE) framework that tackles task scalability by further unlocking the model's parameter scalability. Specifically, we enhance both the architecture and the optimization of the agent, where we strengthen the Decision Transformer (DT) backbone with MoE to reduce task load on parameter subsets, and introduce a three-stage training mechanism to facilitate efficient training with optimal performance. Experimental results show that, by increasing the number of experts, M3DT not only consistently enhances its performance as model expansion on the fixed task numbers, but also exhibits remarkable task scalability, successfully extending to 160 tasks with superior performance.

## 1 INTRODUCTION

Recent developments, such as Decision Transformer (Chen et al., 2021) and Trajectory Transformer (Janner et al., 2021), have reframed offline reinforcement learning (RL) as a sequence modeling problem, showcasing their ability to transform large-scale datasets into potent decision-making agents. These models also prove valuable for multi-task RL (MTRL), offering a high-capacity framework capable of accommodating task variances and assimilating knowledge from diverse datasets. Additionally, they pave the way for incorporating innovations from language modeling (Brown et al., 2020) into MTRL methodologies, unlocking new potential for cross-disciplinary advancements.

Drawing inspiration from large language models (LLM), where models harness a remarkable generalization capability to address a wide range of tasks, there's a growing interest in the potential of training RL agent to master increasingly diverse tasks. However, the application of these high-capacity sequential models to massive multi-task RL presents considerable challenges. Firstly, existing approaches exhibit limited scalability with respect to task numbers. With most studies confined to dozens of tasks in Atari or Meta-World (Lee et al., 2022; He et al., 2023; Hu et al., 2024), when scaled to a larger number of tasks, their performance degrades significantly; while Gato te preed2022generalist extends the research to over 600 tasks, its performance on simulation control tasks remains suboptimal. Secondly, current research either overlooks the impact of parameter scaling, with most studies confined to very small models (Xu et al., 2022), or overly relies on the inherent parameter scalability of the Transformer architecture. Although Gato (Reed et al., 2022) and Multi-Game DT (Lee et al., 2022) have experimentally demonstrated the performance enhancements with expanded model size, a comparison between the increase in model size and the corresponding performance gains suggests that these approaches cannot be considered an efficient method of parameter scaling.

---

*Corresponding Author

To address the above challenges, we extend the research scenario to encompass 160 simulation control tasks. We begin by investigating the pivotal role of task quantity on model performance and gradient conflicts, followed by an exploration of the impact of model expansion on MTRL. With the evident phenomenon that model performance and gradient conflicts deteriorate as the task scales, our findings indicate that these declines are most pronounced when the task number remains relatively low; once the number of tasks reaches sufficiently large, the performance degradation tends to become gradual and steady. Therefore, one of our key insights emerged when viewed from a reverse perspective: *reducing the learning task numbers to a sufficiently small scale can significantly enhance the performance.* In our study on model scaling, the results surprisingly reveal that simply increasing the model size rapidly hits the performance ceiling. Thus, naively expanding shared parameters to offset the performance degradation from an increasing task numbers proves ineffective. In contrast, expanding parameters while reducing the task numbers results in the most pronounced performance gains. So *how can we really minimize the number of tasks to be learned while efficiently scaling model parameters, thereby maximizing performance?*

Building upon these insights, we propose M3DT, a **M**oE-based **DT** for handling **M**assive **M**ulti-task RL. The overall framework is shown in Figure 2. We make improvements in both the architecture and optimization of the RL agent. Specifically, we introduce a mixture-of-experts (MoE) in the DT backbone to achieve parameter separation, allowing the backbone to learn shared knowledge across all tasks, while each expert specializes in learning task-specific knowledge from a distinct small task subset, which greatly simplifies the training of parameters in experts. This also unlocks effective parameter expansion, as the number of experts is the most efficient way to scale models (Fedus et al., 2022). By increasing the number of experts, we can not only introducing a large number of parameters, but reduce the task load on each parameter subset, which mutually reinforce the model performance. Furthermore, M3DT introduces a three-stage training mechanism, sequentially optimize the DT backbone, each expert, and the router, respectively, which allows each module to explicitly learn specialized knowledge without interference, mitigating the severe gradient conflicts encountered when scaling to massive tasks, meanwhile reducing the difficulty of MoE training. Experimental results show that, by increasing the number of experts, on one hand, M3DT can consistently enhance its performance as model expansion on the fixed task scale; on the other hand, it exhibits remarkable task scalability, successfully extending to 160 tasks with superior performance. In summary, our research makes three significant contributions to the field of MTRL:

1. We rethink the challenges of sequence modeling in MTRL from the perspective of task numbers and model size, analyze the performance degradation and gradient conflicts with increasing task numbers, and identify the limited effectiveness of naively parameter scaling in handling massive multi-task scenarios. (Section 2)
2. Based on the above insights, we propose M3DT, a novel framework that enhances the DT architecture with MoE, explicitly assign the grouped task subsets to each expert through task grouping, and introduce a three-stage training mechanism for training each module without interference. By increasing the number of experts, we unlock the parameter scalability for mastering massive tasks. (Section 3)
3. We demonstrate the superior performance of M3DT through rigorous testing on a broad spectrum of task scales, analyze its functionality through extensive ablation studies, and verify its task scalability and parameter scalability. (Section 4)

## 2 RETHINKING DT WITH MTRL

In this section, we delineate two primary challenges of employing DT in MTRL: the inability to scale with task number and model size, laying the groundwork for the motivation behind our method.

### 2.1 LIMITED SCALABILITY OF TASK NUMBERS

The complexity of MTRL is significantly amplified with increasing task numbers, largely attributed to escalating gradient conflicts. These conflicts stem from optimizing a shared set of parameters across tasks with differing objectives, leading to compromises in task-specific optimization. To better understand the impact of task numbers, we use Prompt-DT (Xu et al., 2022) and construct an offline dataset containing 160 tasks based on Meta-World (Yu et al., 2020b), DM Control (Tassa et al., 2018), and Mujoco Locomotion (Todorov et al., 2012), investigating the performance trends as the task number scales from 10 to 160. We measure the model's performance across three benchmarks using normalized scores. The gradient conflict is assessed by the average cosine similarity between the aggregate gradient and the gradients of each task, where a lower similarity indicates a higher degree of gradient conflict. The implementation details are described in Appendix B.

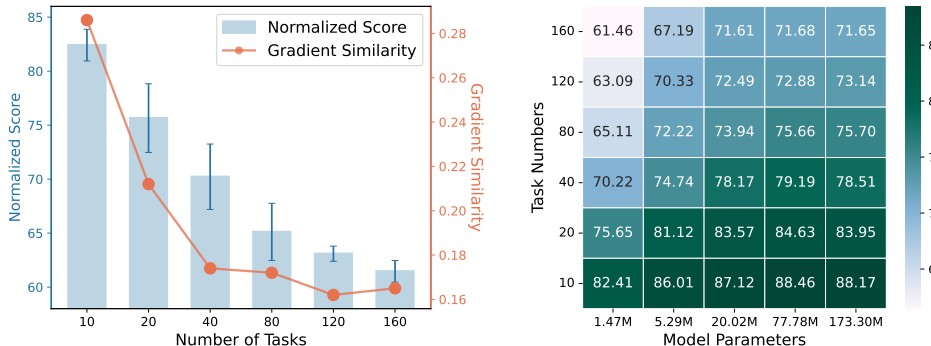

Figure 1: **(left)** With the number of tasks increases from 10 to 160, both model performance and gradient similarity experience a noticeable decline. **(right)** As the model size increases, model performance rapidly reaches its ceiling and ceases to improve thereafter.

During the expansion of the training tasks, we observe a divergent trend in the performance across different task scales. The results presented in Figure 1 (left) highlight three distinct phenomena: • **Normalized Score:** In the clear trend where the performance degrades with increasing task numbers, the decline is pronounced when task number is relatively low (below 40 tasks), while it becomes much more gradual once the tasks reach a sufficiently large number (above 80 tasks). • **Gradient Conflicts:** The decrease of gradient similarity generally aligns with that of model performance, while it drops more rapidly when the task number is low, and, counterintuitively, levels off and shows minimal decline after exceeding 40 tasks. • **Performance Variance:** For each run, we test on different task sets with the same number of tasks, thus the standard deviation of normalized scores reflects the model's robustness to variations in task combinations. For a massive number of tasks (120 to 160), the standard deviation is small, as the variations in task subsets from the total 160 tasks are minimal. For a moderate number of tasks (20 to 80), the standard deviation is large, indicating that different task combinations significantly impact performance, as similar tasks are easier for learning. For few tasks (10), despite greater variability in sampled task subsets, the standard deviation is much smaller than that in moderate tasks, which underscores that when task number is sufficiently low, the learning process is simple enough to be affected by the inter-task relationships.

Thus, a reverse perspective reveals: *reducing the learning task number, particularly to a sufficiently small scale, can significantly enhance model performance.* When the task number is small enough, challenges such as gradient conflicts and inter-task relationships become effortlessly manageable.

## 2.2    LIMITED SCALABILITY OF MODEL SIZE IN MTRL

Since existing models struggle with large-scale tasks in MTRL, a straightforward approach is to scale up the model size to enhance its capacity. The supervised learning community has convincingly demonstrated larger networks lead to improved performance, in particular for language models (Kaplan et al., 2020). To investigate whether this trend holds for the DT model in MTRL, we conduct experiments with Prompt-DT, a language model-based method trained in supervised learning for MTRL, of varying sizes across different task scales. Detailed implementation can be found in Appendix B.4. Results in Figure 1 (right) highlight some of the surprising phenomena, which contradict the behaviors typically observed in previous research (Lee et al., 2022; Reed et al., 2022):

**In MTRL, increasing the model size of DT swiftly hits the performance ceiling, preventing sustained improvements.** The horizontal comparison in Figure 1 (right) shows that expanding the model parameters is effective within a limited range across all task scales; once the model exceeds 20M parameters, further scaling yields no meaningful improvements. This is clearly inefficient, as scalability is a key advantage of the Transformer architecture, and the performance of approaches relying solely on small models becomes increasingly constrained as the number of tasks scales. This leads to another limitation: **Scaling up the model size can *not* effectively mitigate the performance degradation caused by the increase in task numbers.** As shown by the clear decline from bottom-left to top-right in Figure 1 (right) , larger task numbers prevent the model from achieving the strong performance seen with fewer tasks, even with increased parameters.

Despite these discouraging phenomena, the heatmap reveals the most pronounced performance increase from the top-left to the bottom-right. Building on the insights from the previous section, a natural question arises:

> **Key Insight**: How can we effectively minimize the number of tasks to be learned, while efficiently scaling model parameters, thereby maximizing performance?

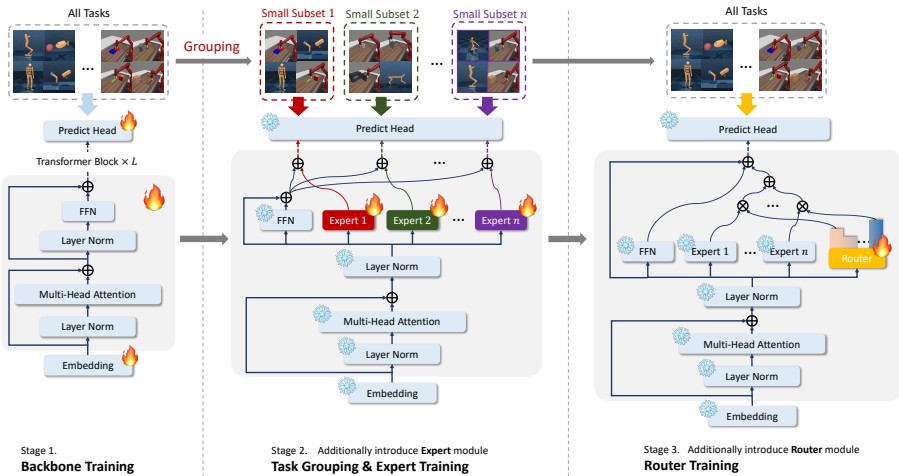

Figure 2: Overview of M3DT. **Stage 1:** We train the PromptDT on all tasks as the backbone. **Stage 2:** We propose task grouping to obtain various small task subsets, and introduce expert module into every transformer block, with each expert handling a specific task subset. Only the experts are optimized in this stage. **Stage 3:** We introduce the router to dynamically assign weights to all experts for training across all tasks. Only the router is optimized in this phase.

## 3 METHODOLOGY: M3DT

From the previous observations and discussions, our key insight is that through task grouping and parameter separation, we can assign much fewer tasks to each parameter subset, leading to exceptional performance. And by expanding the number of parameter subsets while preserving their individual sizes, and simultaneously increasing the number of task groups, we can both reduce the task load on each parameter subset and enhance the model's capacity with more parameters, which enables a scalable solution to handle an increasing number of tasks by expanding model size without compromising performance.

Based on this insight, we introduce M3DT, which includes two key enhancements in both the architecture and optimization of the agent to improve the task and parameter scalability in MTRL. The first enhancement enables efficient parameter separation and expansion to reduce task load, achieved by incorporating an MoE architecture alongside the FFN in Prompt-DT, as detailed in Section 3.1. The second enhancement includes a task grouping process and three distinct training stages that optimize the agent's learning process, especially for ensuring each parameter subset can effectively handle a specific task subset without interference, as outlined in Section 3.2. The overall framework of our method is illustrated in Figure 2.

### 3.1 ARCHITECTURE: MoE FOR REDUCING TASK LOAD

We propose to use an MoE architecture with scalable experts for massive tasks. The MoE structure is characterized by its composition of $N$ modular experts and a router, $\theta_{\mathrm{MoE}} = \{\theta_1, ..., \theta_N, \theta_r\}$, where each expert $\theta_i$ is responsible for a distinct task subset and the router $\theta_r$ dynamically assigns weights for these experts. Accordingly, the parameters of a given expert are updated only by gradients from its corresponding small task subset, fully leveraging the remarkable learning capability for fewer tasks. Meanwhile, by increasing the number of experts, we can further decrease the task load assigned to each expert, thereby effectively alleviating the severe gradient conflicts caused by the overwhelming task numbers and enhancing the performance.

The complete architecture of M3DT is illustrated in the right of Figure 2. We employ the complete Prompt-DT (Xu et al., 2022) as the backbone. As previous works have investigated the crucial role of the feed-forward network (FFN) in transformer module in multi-task settings (Tang et al., 2024), we incorporate the MoE architecture by augmenting each Transformer block with additional experts and the router alongside the preserved FFN, which can maintain the learned shared knowledge from all tasks in the backbone model. We use networks identical to the FFN module as experts and employ an MLP as the router. By this, the MoE and the backbone work in tandem, enabling the agent to dynamically adjust the utilization of different experts for handling diverse types of tasks.

### 3.2 OPTIMIZATION: THREE-STAGE TRAINING MECHANISM

Considering our unique training purpose, namely the explicit task assignment and independent training for each expert, as well as the inherent challenges of training MoE, we propose a three-stage

training mechanism, with distinct training processes designed for the backbone, experts, and router, each trained separately. The overall training process is illustrated in Figure 2.

**Backbone training with minimal gradient conflicts.** We first train Prompt-DT on all tasks to capture shared knowledge, enabling it to embed information and predict actions across tasks, which serves as the backbone architecture for M3DT. This training process results in intense gradient conflicts, as illustrated in Figure 1 (left). To gain deeper insight into the progression of gradient conflicts during training, we conduct experiments that illustrate how performance and gradient conflicts develop across training iterations in Figure 5 (left). Initially, the model's performance improves rapidly with gradient conflicts escalate from a low initial value, enabling fast knowledge acquisition. However, as training progresses, gradient conflicts rapidly reach the peak, leading to diminishing performance gain. Based on this, we restrict training of the backbone to the early stage before gradient conflicts reach their peak. This strategy allows the model to efficiently learn shared knowledge, ensuring the shared parameters align with the solution space while preventing excessive updates on dominant tasks, thereby minimizing negative interference with conflicting tasks. More discussion is detailed in Section C. This approach provides a solid foundation for the subsequent expert training.

**Task grouping and experts individually training.** We employ task grouping and then train the experts on smaller task subsets, which not only reduces the task load to achieve better performance, but also mitigates the severe gradient conflicts that arise in shared parameters after a certain stage of training. We begin by proposing two task grouping methods: (1) random grouping: based on the satisfactory results on few tasks as illustrated in Figure 1 (left), we randomly divide all tasks into equally sized subsets for naively reducing the task number to a lower value; (2) gradient-based grouping: we first calculate the agreement vectors (Hu et al., 2024) (as detailed in Appendix D) for each task in the current backbone as a measure of task similarity, and then apply K-means to group these vectors and the corresponding tasks. After obtaining the task subsets, we introduce a dedicated expert for each subset. The expert module is trained on its specific task subset with the backbone parameters frozen, preserving the shared knowledge while allowing the expert to focus on learning task-specific information. By limiting the size of each task subset, we can ensure effective learning on these parameter subsets. Additionally, training each expert independently helps mitigate the issue of imbalanced updates among experts in MoE. Consequently, when dealing with large-scale tasks, we can expand the number of experts and task groups to mitigate the severe performance drop.

**Router training.** Finally, we train the router on all tasks, enabling it to dynamically assign weights to different experts for various tasks. In this stage, we freeze the parameters of both the backbone and all experts, allowing only the router to be optimized. This strategy significantly simplifies the training of the MoE while preserving the knowledge already acquired by the model.

## 4 EXPERIMENTS AND ANALYSIS

In this section, we conduct extensive experiments to answer the following questions: (1) How does M3DT compare to other baselines in the massive multi-task regime? (2) Does M3DT exhibit task scalability and parameter scalability? (3) What makes M3DT effective? The detailed implementations are illustrated in Appendix B, and further analysis is illustrated in Appendix C due to page limitation.

### 4.1 M3DT HELPS TASK AND PARAMETER SCALABILITY

In this study, we benchmark M3DT and its variants against baselines on different task scales. We compare the performance of the baselines at their default size and an expanded size. The variants of M3DT include **M3DT-Random**, which employ random grouping to obtain task subsets and train experts; **M3DT-Gradient**, which utilizes gradient information for grouping.

As shown in Table 1, M3DT-Random surpasses all other methods at all task scales, achieving a 0.1%, 4.3% and 5.4% improvement in 10, 80 and 160 tasks, respectively, compared to the best baseline. The advantages of our method become more pronounced as the number of tasks increases. By employing random grouping, M3DT-Random already effectively competes with the current state-of-the-art techniques, highlighting the significant effectiveness of introducing parameter subsets and reducing corresponding task load. Furthermore, M3DT-Gradient enhances the performance by identifying better task groups to mitigate learning complexity, resulting in substantial gains of 6.6% and 7.5% in 80 tasks and 160 tasks, respectively. M3DT also effectively alleviates the performance degradation with increasing task numbers. Compared to the severe performance drop of around 20% observed in other baselines when scaling tasks numbers from 10 to 160, our approach only experienced a 12.3% decline. And M3DT achieves a higher score on 160 tasks than other baselines do on 80 tasks, demonstrating remarkable scalability across task numbers.

Table 1: Comparison of M3DT with baselines of varying sizes across three task scales. M3DT consistently outperforms other baselines and achieves remarkable task scalability, which is attributed to its parameter scalability, as shown in Figure 3.

| Task Scale | 10 Tasks | | 80 Tasks | | 160 Tasks | |
|---|---|---|---|---|---|---|
| Method | Score | Parameters | Score | Parameters | Score | Parameters |
| **MTDT-Small** | $82.75 \pm 2.29$ | 1.47M | $66.65 \pm 0.43$ | 1.47M | $59.19 \pm 1.77$ | 1.47M |
| **MTDT-Large** | $88.92 \pm 0.98$ | 173.30M | $74.38 \pm 1.33$ | 173.30M | $70.65 \pm 1.67$ | 173.30M |
| **PromptDT-Small** | $82.41 \pm 1.47$ | 1.47M | $65.11 \pm 2.65$ | 1.47M | $61.46 \pm 1.78$ | 1.47M |
| **PromptDT-Large** | $88.17 \pm 1.41$ | 173.30M | $75.70 \pm 1.62$ | 173.30M | $71.65 \pm 1.13$ | 173.30M |
| **HarmoDT-Small** | $79.75 \pm 1.32$ | 1.47M | $60.71 \pm 3.91$ | 1.47M | $57.27 \pm 0.84$ | 1.47M |
| **HarmoDT-Large** | $86.63 \pm 1.28$ | 173.30M | $75.56 \pm 1.69$ | 173.30M | $72.80 \pm 2.89$ | 173.30M |
| **M3DT-Random** (Ours) | $89.09 \pm 1.20$ | 47.87M | $78.92 \pm 1.21$ | 98.37M | $76.74 \pm 0.94$ | 174.12M |
| **M3DT-Gradient** (Ours) | $\textbf{89.23} \pm 1.12$ | 47.87M | $\textbf{80.66} \pm 0.97$ | 98.37M | $\textbf{78.21} \pm 0.47$ | 174.12M |

The scalability of M3DT in handling massive tasks is attributed to its parameter scalability, which can meanwhile reduce the task load on each parameter subset by increasing the number of experts, thereby enhancing overall performance. We conduct experiments on the number of experts across 80 tasks and 160 tasks, as shown in Figure 3. For 10 tasks, 8 experts already yield excellent results, achieving scores of 89.09 and 89.23 with 47.87M parameters. In 80 and 160 tasks, expanding the parameter size within a certain range through increasing the number of experts can significantly improve

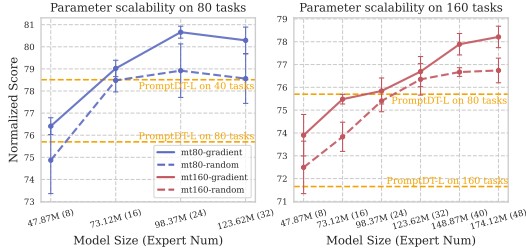

Figure 3: By increasing the number of experts, M3DT effectively unlocks the parameter scalability, further helping to tackle task scalability.

model performance, achieving a 11.2% and 11.7% improvement, separately. Since both M3DT-Random and M3DT-Gradient show diminishing performance gains after reaching 40 experts on 160 tasks, we use 40 experts in subsequent experiments unless stated otherwise. Additionally, we analyze the reasons why increasing the number of experts cannot continually improve performance and eventually reaches a performance ceiling in Appendix C.

## 5 RELATED WORK

**Multi-Task RL** Multi-task RL aims to learn a shared policy for a diverse set of tasks. One of the most straightforward approaches to MTRL is to formulate the multi-task model as task-conditional sequence modeling Xu et al. (2022); Reed et al. (2022); Lee et al. (2022); He et al. (2023); Hu et al. (2024). Some methods also focus on handling gradient conflicts among different tasks Yu et al. (2020a); Chen et al. (2020); Liu et al. (2021). On the other hand, some methods employed a dedicated shared structure to leverage the shared knowledge D'Eramo et al. (2024); Yang et al. (2020); Sun et al. (2022). While most of these methods address tens or even less tasks, we focus on scaling to a significantly larger number of tasks, achieving task scalability.

**Mixture-of-Experts** MoEs have recently helped scaling language models up to trillions of parameters thanks to their modular nature Lepikhin et al. (2020); Fedus et al. (2022). MoEs also help performance in multi-task settings Fan et al. (2022); Ye & Xu (2023); Dou et al. (2024). There have been few works exploring MoEs in RL for single Akrour et al. (2021); Obando-Ceron et al. (2024) and multi-task learning Hendawy et al. (2023); Huang et al. (2024). While M3DT explicitly assign tasks to specific experts and train each expert independently to handle multi-task problems.

## 6 CONCLUSION

In this study, we first delve into the challenges of sequence modeling in MTRL, analyze the performance degradation and gradient conflicts with increasing task numbers, and identify the limited effectiveness of naively scaling parameter in MTRL. Based on these insights, we introduce M3DT, a novel approach designed to unlock the parameter scalability for handling massive tasks. By employing task grouping and MoE, M3DT significantly reduces the task load assigned to each parameter subset while enhancing the overall performance through the expansion of experts. We propose a three-stage training mechanism that allows explicit task assignment to each expert, enabling sequential training of different modules without interference. Our empirical evaluations across diverse task scales underscore M3DT's superior performance compared to existing baselines, establishing its state-of-the-art effectiveness in MTRL scenarios.

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

## A  PRELIMINARY

**Offline Reinforcement Learning.** The goal of RL is to learn a policy $\pi_\theta(a|s)$ maximizing the expected return $\mathbb{E}[\sum_{t=0}^{\infty} \gamma^t \mathcal{R}(s_t, a_t)]$ in a Markov Decision Process (MDP) $(\mathcal{S}, \mathcal{A}, \mathcal{P}, \mathcal{R}, \gamma, d_0)$, with state space $\mathcal{S}$, action space $\mathcal{A}$, environment dynamics $\mathcal{P}(s'|s,a) : \mathcal{S} \times \mathcal{S} \times \mathcal{A} \rightarrow [0,1]$, reward function $\mathcal{R} : \mathcal{S} \times \mathcal{A} \rightarrow \mathbb{R}$, discount factor $\gamma \in [0,1)$, and initial state distribution $d_0$ (Sutton & Barto, 2018). In the offline setting (Levine et al., 2020), a static dataset $\mathcal{D} = \{(s, a, s', r)\}$, collected by a behavior policy $\pi_\beta$, is provided. Offline RL algorithms learn a policy entirely from this static offline dataset, without any online interactions with the environment.

**Multi-Task RL.** In multi-task RL, different tasks can have different reward functuions, state spaces, and transition functions. Given a specific task $\mathcal{T} \sim p(\mathcal{T})$, a task-specified MDP can be defined as $(\mathcal{S}^\mathcal{T}, \mathcal{A}^\mathcal{T}, \mathcal{P}^\mathcal{T}, \mathcal{R}^\mathcal{T}, \gamma, d_0^\mathcal{T})$. Instead of solving a single MDP, the goal is to find an optimal policy that maximizes expected return over all tasks: $\pi^* = \arg\max_\pi \mathbb{E}_{\mathcal{T} \sim p(\mathcal{T})} \mathbb{E}_{a_t \sim \pi}[\sum_{t=0}^{\infty} \gamma^t r_t^\mathcal{T}]$. The static dataset $\mathcal{D}$ is correspondingly partitioned into per-task sub-sets as $\mathcal{D} = \cup_{i=1}^N \mathcal{D}_i$, where $N$ is the number of tasks.

**Prompt Decision Transformer.** The integration of Transforme (Vaswani, 2017) in offline RL for sequence modeling has gained prominence in recent years, such as Decision Transformer (Chen et al., 2021). Prompt-DT (Xu et al., 2022) extends DT by using task-specific prompts to enhance multi-task learning and few-shot generalization. Unlike text-based prompt in NLP (Liu et al., 2023), Prompt-DT employs short trajectories as prompts, which consist of state, action, and return-to-go tuples $(s^*, a^*, \hat{r}^*)$, providing directed guidance to RL agents with few-shot demonstrations. Each element marked with the superscript $\cdot*$ is relevant to the trajectory prompt. These trajectory prompts are much shorter than the task's horizon, encompassing essential information to only identify task, yet inadequate for task imitation. During training with offline data, Prompt-DT utilizes $\tau_{i,t}^{input} = (\tau_i^*, \tau_{i,t})$ as input for each task $\mathcal{T}_i$, combining a $K^*$-step trajectory prompt $\tau_i^*$ with a normal $K$-step trajectory $\tau_{i,t}$. The prompt trajectory is formulated as:

$$\tau_{i,t}^{input} = (\hat{r}_{i,1}^*, s_{i,1}^*, a_{i,1}^*, ..., \hat{r}_{i,K^*}^*, s_{i,K^*}^*, a_{i,K^*}^*,$$
$$\hat{r}_{i,t-K+1}, s_{i,t-K+1}, a_{i,t-K+1}, ..., \hat{r}_{i,t}, s_{i,t}, a_{i,t}). \tag{1}$$

The action $a$ is predicted through a prediction head linked to the state token. The training objective aims to minimize the mean-squared loss:

$$\mathcal{L}_{DT} = \mathbb{E}_{\tau_{i,t}^{input} \sim \mathcal{D}_i} \left[ \frac{1}{K} \sum_{m=t-K+1}^{t} (a_{i,m} - \pi(\tau_i^*, \tau_{i,m}))^2 \right]. \tag{2}$$

Prompt-DT has become a widely used backbone in recent research on MTRL.

## B  EXPERIMENTAL DETAILS

### B.1  DETAILED ENVIRONMENTS

We consider a total of 160 continuous control tasks from 3 task domains: Meta-World Yu et al. (2020b), DMControl Tassa et al. (2018), Mujoco Locomotion Todorov et al. (2012). This section provides an exhaustive introduction of the tasks considered, including their observation and action dimensions, and the calculation of normalized score. Our goal is not to propose a new benchmark with 160 tasks, but rather to use a sufficiently large number of tasks to explore the impact of task quantity.

### B.1.1 META-WORLD

The Meta-World benchmark encompasses a diverse array of 50 distinct manipulation tasks, unified by shared dynamics. These tasks involve a Sawyer robot engaging with a variety of objects, each distinguished by unique shapes, joints, and connective properties. The complexity of this benchmark lies in the heterogeneity of the state spaces and reward functions across tasks, as the robot is required to manipulate different objects towards varying objectives. The robot operates with a 4-dimensional fine-grained action input at each timestep, which controls the 3D positional movements of its end effector and modulates the gripper's openness. The state space is unified into 39 dimension. In its original configuration, the Meta-World environment is set with fixed goals, a format that somewhat limits the scope and realism of robotic learning applications. To address this and align with recent advancements in the field, as noted in works by Yang et al. (2020); Sun et al. (2022); He et al. (2023); Hu et al. (2024), we have modified all tasks to incorporate a random-goal setting. For the offline dataset, we follow the works He et al. (2023); Hu et al. (2024) and utilize their dataset with the near-optimal trajectories, which consists of the experience from random to expert (convergence) in SAC-Replay Haarnoja et al. (2018). The primary metric for evaluating performance in this benchmark is the average success rate across all tasks, providing a comprehensive measure of the robotic system's adaptability and proficiency in varied task environments. We directly use the success rate on each task as its normalized score.

### B.1.2 DMCONTROL

The tasks in DMControl involve significantly more diverse embodiments, state spaces, action spaces, and reward functions, which greatly increases their complexity. We consider a total of 30 continuous control tasks in the DMControl domain, including 19 original DMControl tasks and 11 new (custom) tasks created specifically for M3DT benchmarking, following the work Hansen et al. (2023). We directly use the dataset collected by Hansen et al. (2023), and we only use the first 2,000 trajectories for each task to ensure consistency in dataset size with other tasks. We list all used DMControl tasks in Table 2. For evaluation, we linearly scale the original reward range of [0,1000] to [0,100], using it as our normalized score.

### B.1.3 MUJOCO LOCOMOTION

In this paper, we also employ a diverse array of meta-RL control tasks to construct a dataset with a sufficient number of tasks for exploring the challenges of MTRL when confronted with a large number of tasks. We directly utilize the datasets proposed by Xu et al. (2022). The tasks are detailed as follows:

- **Cheetah-vel**: It defines 40 unique tasks, each associated with a specific goal velocity, uniformly distributed between 0 and 3 m/s. The agent's performance is assessed based on the l2 error relative to the target velocity, with a penalty for deviations. These 40 tasks share a unified state space of 20 and an action space of 6. Based on the reward ranges of these environments, we linearly map the return values within the interval [-100, -30] to the normalized range of [0, 100] as our normalized scores, while returns outside this range are directly capped at 0 or 100, respectively.

- **Ant-dir**: We also use 40 tasks in this domain, each with a goal direction uniformly sampled in a two-dimensional plane. The agent, an 8-jointed ant, is incentivized to attain high velocity in the designated direction. The state space for these tasks has a dimensionality of 27, and the action space consists of 8 dimensions. We linearly map the return values within the interval [0, 500] to the normalized range of [0, 100], while returns outside this range are directly capped at 0 or 100, respectively, to calculate our normalized scores.

By using normalized scores, we can align tasks with initially inconsistent evaluation metrics, enabling us to assess the model's ability to simultaneously tackle multiple tasks. To address the inconsistency in state and action spaces across tasks, we zero-pad all states and actions to their largest respective dimensions (i.e. 39 and 8, respectively), and mask out invalid action dimensions in predictions made by the policy during both training and inference.

### B.2 BASELINES

We compare our proposed M3DT with the following DT-based baselines.

Table 2: DMControl tasks used in this paper.

| Task | Observation dim | Action dim | New? |
|------|-----------------|------------|------|
| Acrobot Swingup | 6 | 1 | N |
| Cartpole Balance | 5 | 1 | N |
| Cartpole Balance Sparse | 5 | 1 | N |
| Cartpole Swingup | 5 | 1 | N |
| Cartpole Swingup Sparse | 5 | 1 | N |
| Cheetah Jump | 17 | 6 | Y |
| Cheetah Run | 17 | 6 | N |
| Cheetah Run Back | 17 | 6 | Y |
| Cheetah Run Backwards | 17 | 6 | Y |
| Cheetah Run Front | 17 | 6 | Y |
| Cup Catch | 8 | 2 | N |
| Cup Spin | 8 | 2 | Y |
| Finger Spin | 9 | 2 | N |
| Finger Turn Easy | 12 | 2 | N |
| Finger Turn Hard | 12 | 2 | N |
| Fish Swim | 24 | 5 | N |
| Hopper Hop | 15 | 4 | N |
| Hopper Hop Backwards | 15 | 4 | Y |
| Hopper Stand | 15 | 4 | N |
| Pendulum Spin | 3 | 1 | Y |
| Pendulum Swingup | 3 | 1 | N |
| Reacher Easy | 6 | 2 | N |
| Reacher Hard | 6 | 2 | N |
| Reacher Three Easy | 8 | 3 | Y |
| Reacher Three Hard | 8 | 3 | Y |
| Walker Run | 24 | 6 | N |
| Walker Run Backwards | 24 | 6 | Y |
| Walker Stand | 24 | 6 | N |
| Walker Walk | 24 | 6 | N |
| Walker Walk Backwards | 24 | 6 | Y |

- **MTDT**: We extend the DT architecture Chen et al. (2021) to learn from multitask data. Specifically, MTDT concatenates an embedding $z$ and a state $s$ as the input tokens, where $z$ is the encoding of task ID. In evaluation, the reward-to-go and task ID are fed into the Transformer to provide task-specific information. Leveraging the scalability of the Transformer architecture, we compare the performance of this method at both its default size (1.47M) and expanded size (173.30M).

- **PromptDT** Xu et al. (2022): PromptDT built on DT aims to learn from multi-task data and generalize the policy to unseen tasks. It leverages short task trajectories as prompts to guide the model in identifying the current task. PromptDT generates actions based on the trajectory prompts and reward-to-go. We compare the performance of this method at both its default size (1.47M) and expanded size (173.30M).

- **HarmoDT** Hu et al. (2024): HarmoDT built on PromptDT aims to lean a specific mask for each task, effectively shielding the model's parameters that conflict most with the task, thereby mitigating the severe gradient conflicts in MTRL. During evaluation, HarmoDT also generates actions based on the trajectory prompts and reward-to-go. We compare the performance of this method at both its default size (1.47M) and expanded size (173.30M).

## B.3 TASK SELECTION

To mitigate the impact of task selection and combination on performance, we use different task combinations for each run seed in scenarios involving 10, 20, 40, 80, and 120 tasks. To fairly analyze the effects solely caused by task quantity, we maintain consistent average task difficulty across various task scales and seeds. Specifically, we train 160 separate PromptDT models for each of the 160 tasks and compute the score for each task, which serves as the task difficulty for that task. When selecting tasks for different task scales, we ensure that the average task difficulty of the chosen task set aligns

with that of the complete set of 160 tasks. We use the 1.47M PromptDT model to compute the task difficulty and derive the task sets for all task scales. Subsequently, all experiments are conducted using the same task set to eliminate potential bias.

## B.4 IMPLEMENTATION OF MODEL EXPANSION

In this experiment, we do not invest significant effort into exploring the optimal structure for each baseline model at different parameter scales. Instead, we adopt their default number of layers and attention heads, only increasing the width of the models, i.e. the dimension of their hidden states. Although previous research Kaplan et al. (2020) have shown that the model structure has only a marginal impact on performance when the parameter size remains constant, we conduct experiments with PromptDT to verify whether our expanding approach is reasonable in the context of MTRL. The results are shown in Figure 4, where $d_{model}$ denotes the width of the model, $n_{layer}$ and $n_{head}$ denote the number of layers and attention heads, respectively. In MTRL, the model performance is also weakly depends on the model architecture. We summarize the specific structures of each scaled model in Table 3.

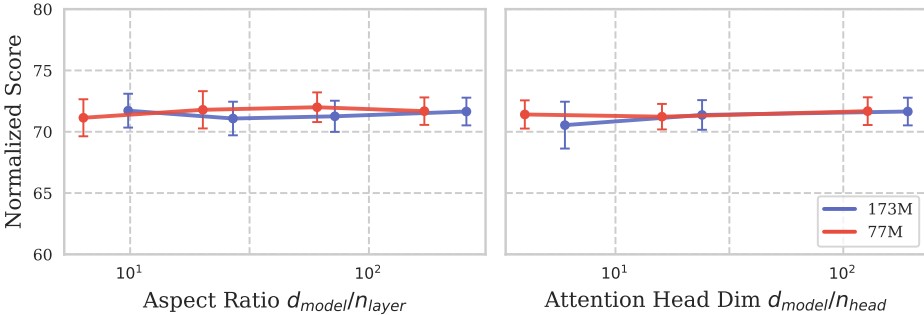

Figure 4: Performance depends very mildly on model shape when the total number of parameters is held fixed.

Table 3: DMControl tasks used in this paper.

| Parameters | Layers | Attention Heads | Model Width |
|------------|--------|-----------------|-------------|
| 1.47M | 6 | 8 | 128 |
| 5.29M | 6 | 8 | 256 |
| 20.02M | 6 | 8 | 512 |
| 77.78M | 6 | 8 | 1024 |
| 173.30M | 6 | 8 | 1536 |

## B.5 EXPERIMENTAL SETUPS, HYPER-PARAMETERS AND RESOURCES

In this section, we introduce the implementation for M3DT. We employ the PromptDT with 5.29M parameters as the backbone of M3DT, with the structure outlined in Table 3. The structures of our introduced experts are identical to that of the FFN in the backbone. We employ a 5-MLP as our router. The padded input and output dimensions are 39 and 8, respectively, as illustrated in B.1. All experiment in this paper are run with 3 seeds. The specific model parameters and hyper-parameters utilized in our training process are outlined in Table 4. We use NVIDIA GeForce RTX 4090 to train and evaluate each model except HarmoDT-Large, while it is trained and evaluated on NVIDIA A100 40G due to its substantial resource requirements.

Table 4: Hyper-parameters of M3DT in our experiments.

| Parameter | Value |
|---|---|
| Number of layers | 6 |
| Number of attention heads | 8 |
| Hidden dimension | 256 |
| Number of experts | [8,16,24,32,40,48] |
| Nonlinearity function | ReLU |
| Batch size | 16 |
| Prompt length $K$ | 20 |
| Dropout | 0.1 |
| Learning rate | 1.0e-4 |
| Optimizer | Adam |
| Total rounds | 1e6 |
| -Backbone training rounds | 4e5 |
| -Expert training rounds | 2e5 |
| -Router training rounds | 4e5 |

## C  FURTHER ANALYSIS

### C.1  IS GROUPED TRAINING IMPORTANT?

To investigate whether the success of M3DT is primarily due to the explicit task grouping and three-stage training mechanism, or the inherent advantages of the MoE architecture itself, we conduct a detailed ablation study, as illustrated in Table 5. Despite utilizing the MoE structure, the absence of our meticulously designed training strategy leads to a substantial performance degradation. Training a PromptDT with MoE end-to-end from scratch (i.e. M3DT w/o 3-stage training) only yields results similar to those of a standalone PromptDT with the same parameter scale, while freezing the trained backbone and jointly training all experts and the router on all tasks (i.e. M3DT w/o grouping) results in a worse performance, with a score of 67.34. In addition, after training the experts in groups, simultaneously fine-tuning them when training the router (i.e. both M3DT-R/G w/o expert freezing) also leads to suboptimal results. This further underscores the validity of our entire framework.

Table 5: Ablation study on different training process of M3DT, which illustrate the effectiveness of our dedicated three-stage training mechanism.

| Method | Normalized Score |
|---|---|
| M3DT-R | $76.67 \pm 0.29$ |
| M3DT-G | $77.89 \pm 0.47$ |
| M3DT w/o 3-stage training | $71.90 \pm 0.70$ |
| M3DT w/o grouping | $67.34 \pm 0.56$ |
| M3DT-R w/o expert freezing | $71.89 \pm 0.63$ |
| M3DT-G w/o expert freezing | $71.88 \pm 0.62$ |

### C.2  IS EARLY STOPPING OF BACKBONE TRAINING TRULY EFFECTIVE?

As illustrated in Figure 5 (left), the gradient conflicts in the PromptDT backbone progressively escalate with training duration, eventually reaching a peak, after which the performance improvements become notably sluggish. To assess the efficacy of early stopping, which is employed to mitigate these severe gradient conflicts at the expense of some performance gains, we conduct a comparison of M3DT using PromptDT trained for varying steps as backbone. The results are depicted in Figure 5 (right). M3DT consistently achieves strong performance across various training durations, with early stopping at 400k steps, just as gradient conflicts peak, yielding the most optimal results. This demonstrates the robustness of M3DT in the initial training phase. Training for only 200k steps prevents the backbone from fully learning the shared knowledge, causing the parameters to deviate from the optimal solution space, resulting in suboptimal performance. On the other hand, continuing training beyond 400k steps, when gradient conflicts are extremely severe, also leads to a decline in M3DT's performance.

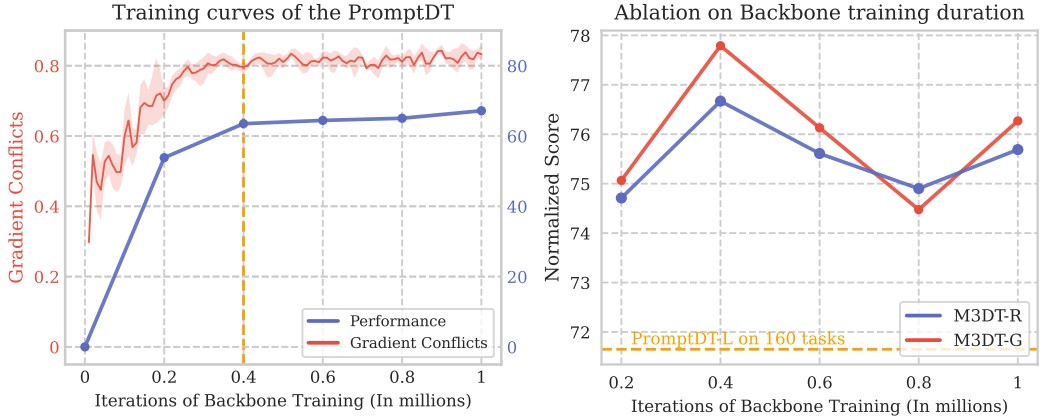

Figure 5: (left) Training curves of the PromptDT, where the early stage exhibits mild gradient conflicts with swift performance gain; (right) The overall performance of M3DT varies with the number of training steps applied to the backbone, with optimal performance occurring just as gradient conflicts reach the peak.

This is likely due to the parameters become overfitted to tasks whose gradients dominate, causing them to deviate further from the solution space of other conflicting tasks. Introducing expert modules at this time for grouped task learning also falls victim to this issue, as they unable to learn the specific knowledge on these conflicting tasks, thus failing to alleviate the performance drop.

## C.3 WHAT AFFECTS PARAMETER EXPANSION?

In this study, we introduce: **Expert Performance:** the averaged results of directly testing all experts on their corresponding task subsets without involving the router, as the upper-bound performance of M3DT; **Small:** a scaled-down version of M3DT, where the width of backbone, experts, and router are all proportionally reduces. As shown in Figure 6, the main factors influencing the parameter scalability of M3DT are threefold: (1) Model width: With the number of experts increasing, M3DT-Small achieves significant performance improvements with modest growth in parameters. However, due to the constrains of the small model width on the capacity

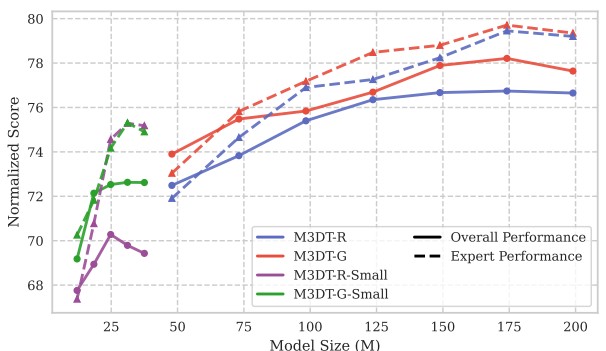

Figure 6: Performance curves of M3DT with increasing model size (i.e., the number of experts) across different base sizes, where each point denotes the addition of 8 experts.

of each module, its scalability to the number of experts is poor, reaching the performance ceiling at only 24 experts. As a result, its parameter scalability is inherently constrained. (2) Router: The performance gap between the dashed and solid lines reflects the performance loss attributed to the router. When the number of experts is small, the router can easily allocate weights across the few experts, resulting in minimal performance loss or even better results. As the number of experts increases, the difficulty of assigning weights to the experts grows, resulting in a progressively larger performance gap, which peaks when the expert performance continues to improve while the overall performance plateaus. (3) Backbone + Expert: Expert performance also tends to plateau when the number of experts becomes sufficiently large. This is primarily due to the shared knowledge learned by the backbone across all tasks is limited, which restricts further performance improvements, regardless of the expert's capabilities. Additionally, when the number of experts is large enough and each expert already faces a sufficiently small task subset, further increasing the number of experts yields diminishing returns in reducing the task load.

## C.4    ROUTER DESIGN.

We compare the Top-K routing (Shazeer et al., 2017) strategy to investigate whether reducing the number of activated experts can enhance the scalability of expert number.    In this experiment, we employed Top-4 routing, and the results are presented in Figure 7. Although our proposed three-stage training mechanism significantly simplifies the training of MoE, Top-4 router fails to scale the number of experts, and performance deteriorates as expert number increases. This is attributed to improper routing load balancing, where certain routes are excessively optimized, combined with the instability in router training induced by sparsity. This finding is consistent with prior work demonstrating the difficulty in scaling up deep RL networks with Top-K router (Obando-Ceron et al., 2024). While it is possible some losses (Riquelme et al., 2021; Mustafa et al., 2022) may result in better Top-K performance, this finding suggests that M3DT benefits from having a weighted combination of all experts from all task subsets.

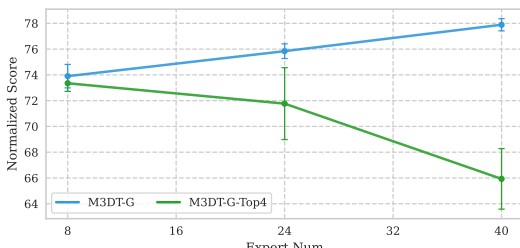

Figure 7: Top-K routing selects the router's top-k outputs, applies softmax to obtain probabilities, and computes the weighted sum of the corresponding expert outputs. M3DT with Top-4 router fails to scale with the number of experts.

## C.5    EXPERT DESIGN.

Our proposed MoE architecture replaces the FFN with an MoE in each transformer block. This is based on what is common practice when adding MoEs to transformer architectures, but is by no means the only way to utilize MoEs. Here we investigate a variant: **Big:** Each expert is a full transformer architecture, where the embedding layer and prediction layer are shared in the backbone. However, M3DT-Gradient-Big only results in a normalized score of 76.53, compared to M3DT-Random scored 77.89, which confirms our intuition that employing normal MoE performs better.

## D    AGREEMENT VECTOR

This section elucidates the utilization of the Agreement Vector, as proposed by Hu et al. (2024), as a metric for task grouping. For each task $\mathcal{T}_i$, the agreement score vector is defined as follows: $A(\mathcal{T}_i) = g_i \odot \frac{1}{N} \sum_{i=1}^{N} g_i$, where $g_i$ denotes the gradient of the parameters calculated from task $\mathcal{T}_i$ and $N$ denotes the total number of tasks. This vector reflects the gradient similarity between the task-specific and the average gradients, and further indicate the task similarity.

