# OpenReview forum: "Mastering Massive Multi-Task Reinforcement Learning via Mixture-of-Expert Decision Transformer"
_ICLR.cc/2025/Workshop/MCDC — MCDC @ ICLR 2025_

### Official Review · Reviewer_fBMt · 2025-02-26

**Rating:** 7
**Confidence:** 3
**Fit:** 4

**Summary:**

Based on their observation that multi-task scaling is challenging for monolithic (decision) transformer architecture, the authors propose a mixture-of-expert model and a three-stage training. The model has both shared parameters as well as parameters that are specific to groups of tasks (experts) and a router, at each transformer block, decides which experts to use based on the incoming representation.

The authors empirically demonstrate the gains of their approach through a primary study and a few additional (ablation studies) in the appendix. The proposed approach seems to scale better to more tasks (up to 160).

**Reason For Giving A Higher Score:**

I am not an expert on recent multi-task RL work so that I might have missed some important relevant literature. This uncertainty prevents me from giving it a higher rating. I also find the paper still has some areas for improvement, even though I realize the authors only have 6-pages.

**Reason For Giving A Lower Score:**

I think this is a solid contribution that is on topic, so I suggest it should be accepted.

**Strengths And Weaknesses:**

The idea of enabling a single model to solve many related tasks in an RL setting is natural and worthwhile exploring. As the authors discuss, much prior work has only studied a relatively modest number of tasks. In that sense, the author's exploration and development push the community forward. I find this to be adequate, especially for a workshop submission.

Regarding weaknesses, I did not find anything major. Overall, the 6-page format limits a bit the type of exploration that might yield further insights. Further unpacking the results in Figure 1 could provide additional insights. I made one or two suggestions about it below. I also did not find the precise definition of the router (in addition to the depiction in Figure 2) and whether it's essential in the setup (more on that below).

**Suggestions:**

- The paper is relatively easy to understand, but there are typos and sentence formulations that could be improved.
- Relatedly, I wasn't sure about the following. Is it really an insight as it's phrased as a question? Also, did you mean to write *maximize* the number of tasks to be learned? "Key Insight: How can we effectively minimize the number of tasks to be learned, while efficiently scaling model parameters, thereby maximizing performance?"

- In Figure 1 (left), I don't precisely understand why gradient similarity goes down since the set of tasks is the same across experiments.
- Gradient conflict. The authors seemingly use conflict and similarity to denote the same concept. Previous work (e.g., [1]) defines conflict as gradients going in opposite directions (i.e., negative cosine). With that in mind, it might be worth separating conflicting gradients from "less similar" gradients to gain additional insights. Other authors (e.g., [2]) have also argued that the gradient magnitude has value. Again, I suggest it might be worth it for the authors to explore/discuss this in their context.
- I didn't understand why the random grouping did so well. You mention "task load," but what does that mean precisely? This relates to my comment above about why gradient similarity is so high when you have only 10 tasks. Perhaps the model can learn a representation that is useful for all tasks.

- I wasn't sure of the router's purpose given the current setup (tasks are observable, and the mapping from expert to task is fixed a priori). That might constitute an interesting ablation study.

- In terms of baselines, it could be interesting to compare against an approach with no parameter sharing (i.e., every model is trained on a single task to learn whether a multi-task setup helps overall and on which tasks it is most (un-)helpful. I don't think this is in the appendix either, but sorry if I missed it.

- At the end of Section 4, you mention "later experiments" without any pointers (maybe you mean Fig 6 in the appendix?).

- Looking at Figure 3, I wonder if you have hypotheses to explain the plateau-ing effect of M3DT as you increase the number of tasks. I am interested in the impact of other hyperparameters (including expanding the width of the backbone and using a different clustering algorithm). I know that some of those were explored in the appendix.

[1] Gradient Surgery for Multi-Task Learning, Yu et al., NeurIPS'20
[2] Task-agnostic continual reinforcement learning: Gaining insights and overcoming challenges, Caccia et al., Collas'23

---

### Official Review · Reviewer_fY6D · 2025-02-28

**Rating:** 7
**Confidence:** 3
**Fit:** 4

**Summary:**

In this paper the authors propose M3DT, a mixture-of-experts (MoE) framework that tackles task scalability by introducing a mixture-of-experts (MoE) module that given a task picks the right expert to execute the task. Authors claim to strengthen the Decision Transformer (DT) backbone with MoE to reduce task load on parameter subsets, and introduce a three-stage training mechanism to facilitate efficient training with optimal performance. Experiments show M3DT’s superior performance compared to existing baselines, establishing its
state-of-the-art effectiveness in MTRL scenarios.

**Reason For Giving A Higher Score:**

The paper fits well with the theme of this conference where they show that simply scaling the model architecture does not help with mastering variety of tasks. They found the root cause of why simple scaling doesn't work, i.e., gradient-conflict, address that issue by a MoE routing technique, and show its effectiveness via experiments. The paper is well written, easy to follow, and backed by experiments.

**Reason For Giving A Lower Score:**

N/A

**Strengths And Weaknesses:**

Strengths: The authors pick a simple idea of mixture of experts to address the gradient-conflict issue in MTRL. The idea although simple works well in practice. They present a 3-step approach of training the M3DT architecture where they first train a common DT on all tasks and then iteratively fine tune each expert on subset of tasks followed by training only the router for effective routing. Combined together this approach shows effective improvement over baselines. The approach is presented neatly in the paper and is easy to follow. The experiments are diverse and extensive enough to convince me about the effectiveness of the approach.

Weaknesses: I'd like to see more detailed study of related work. I found the section to be a bit lacking in coverage. Given the simplicity of the approach I'd like to see how it compares against more SOTA algorithms in MTRL setup that do not rely on DTs. All the baselines discussed in paper are DT-based.

**Suggestions:**

Mentioned in weaknesses.

---

### Official Review · Reviewer_tKxd · 2025-03-03

**Rating:** 7
**Confidence:** 3
**Fit:** 4

**Summary:**

The paper tackles the challenge of scaling multi-task reinforcement learning (MTRL) to a large number of tasks. It first analyzes how performance deteriorates and gradient conflicts arise as task count increases, showing that simply enlarging shared network parameters reaches a performance plateau. To address this, the authors introduce M3DT, which integrates a mixture-of-experts (MoE) architecture within a Decision Transformer (DT) backbone. Using a routing network, it distributes tasks across multiple experts, reducing the effective task load per parameter subset. Experiments across 160 continuous control tasks demonstrate that M3DT mitigates performance degradation as task numbers grow. Ablation studies highlight the impact of explicit task grouping, a staged training strategy, and design choices such as early stopping on the backbone. The paper finally argues that modular parameter separation enables effective scaling and resolves gradient conflicts in large-scale MTRL.

**Reason For Giving A Higher Score:**

The novel architecture combined with extensive experiments (though simulations) make it an interesting contribution that forms a strong basis for future research.

**Reason For Giving A Lower Score:**

The architecture and method might be complex to tune due to numerous components and training stages, and its scalability and real-world applicability are unclear. These should be clarified in the revision.

**Strengths And Weaknesses:**

Strengths:
1. The paper addresses a critical limitation of current multi-task RL approaches - scalability to a large number of tasks.
2. The paper introduces a novel combination of MoE and decision transformer architectures tailored for MTRL.
3. Extensive experiments on 160 tasks across multiple benchmark domains and detailed ablation studies support the claimed improvements over existing methods.
4. The proposed three-stage training method effectively leverages modularity to reduce inter-task interference.

Weaknesses:
1. MoE models can be computationally expensive, especially when scaling the number of experts. The paper does not discuss potential efficiency bottlenecks at large scales.
2. The three-stage training procedure and a complex architecture introduce numerous hyperparameters. The performance might be sensitive to them and better discussions on their tuning and impact would be helpful.

**Suggestions:**

1. Provide more comprehensive ablation studies on hyperparameter sensitivity
2. Consider experiments on additional benchmarks or real-world applications to demonstrate broader applicability.
3. Provide a more detailed discussion regarding the computational cost and training time overhead introduced by the MoE components and multi-stage training.

---

### Official Review · Reviewer_GWAP · 2025-03-04

**Rating:** 8
**Confidence:** 4
**Fit:** 4

**Summary:**

This paper presents a novel framework for multi-task reinforcement learning (MTRL) which first trains a Prompt-DT backbone on all tasks at once for a limited amount of steps, then adds expert modules to the model which are individually trained on subsets of the tasks with the backbone kept frozen and finally trains a router on all tasks to dynamically use the different experts in a weighted fashion. Their framework scales to a very large number of tasks (160) and outperforms considered baselines for 10, 80 and 160 tasks.

**Reason For Giving A Higher Score:**

While the method is novel for the setting of interest, the elements comprising it are not necessarily novel individually.

**Reason For Giving A Lower Score:**

The method is well motivated, obtains good results and adequate ablations are conducted. The paper is fairly well written.

**Strengths And Weaknesses:**

**Strengths:**
- The paper presents an innovative solution to the number of tasks scaling issues encountered in MTRL.
- As far as I can tell the method is novel and the results are strong.
- Extensive experiments on diverse benchmarks (Meta-World, DMControl, Mujoco) and detailed ablation studies reinforce the empirical claims.
- The three-stage training mechanism is well-motivated and addresses the inherent challenges in MoE training.

**Weaknesses:**
- The three-stage training process, while effective, introduces additional complexity and computational overhead. I believe the method is still worth presenting even with the added complexity and computational requirements but for a full paper it would be good to discuss this overhead to increase the methods usefulness for practicians.
- The abstract and the introduction could benefit from a bit of polishing, there are a couple of typos (e.g. citation problem on line 43) and "number of tasks" is more accurate than "task numbers".
- The number of tasks and parameters scaling issues identified don't seem novel, it might be useful to go quicker over those results and more in detail in the rest of the paper.
- Citing more related works from the MoE or "routing among experts" literature would be good. (e.g. (Learning to Route Among Specialized Experts for Zero-Shot Generalization)[https://arxiv.org/abs/2402.05859] or (Towards Modular LLMs by Building and Reusing a Library of LoRAs)[https://arxiv.org/abs/2405.11157]
- More ablations regarding the different training stages would be interesting. Does it help to train everything together at the end? etc.

**Suggestions:**

See weaknesses above.

---

### Decision · Program_Chairs · 2025-03-06

**Decision:**

Accept

**Comment:**

This work proposes a new Mixture-of-Experts system for RL that displays superior task scalability. Being able to learn many tasks is critical for continual collaborative learning and relevant for this workshop. All reviewers recommend acceptance and we're happy to accept it to the workshop.